# Laws for health and care worker protection and rights: A study of 182 countries

**Matthew M. Kavanagh**[1,2]*, **Adi Radakrishnan**[2], **Vishakh Unnikrishnan**[2], **Giorgio Cometto**[3], **Catherine Kane**[3], **Eric A. Friedman**[2], **Varsha Srivatsan**[2], **Luis Gil Abinader**[2], **James Campbell**[3], on behalf of the Health & Care Worker Policy Lab[¶]

1 Department of Global Health, Georgetown University School of Health, Washington, DC, United States of America, 2 Center for Global Health Policy & Politics, O'Neill Institute for National and Global Health Law, Georgetown University Law Center, Washington, DC, United States of America, 3 Health Workforce Department, World Health Organization, Geneva, Switzerland

¶ Membership of the Health & Care Worker Policy Lab is provided in S1 Text.
* matthew.kavanagh@georgetown.edu

**Data Availability Statement:** Data for replication available at Kavanagh, Matthew. 2024. "Replication Data for 'Laws for Health and Care Worker

## Abstract

The unprecedented and multi-faceted challenges health and care workers faced during the COVID-19 pandemic inspired the world's health ministers to call for a new Global Health and Care Worker Compact at the 74[th] World Health Assembly in 2021. The Care Compact identifies key areas where governments can use law and policy to prevent harm, provide support, ensure inclusivity, and safeguard rights of health and care workers toward improving population health. Using policy surveillance methods, we conducted an empirical analysis of the national law and policy environments on health and care workers' protection and rights in 182 countries. Across 10 indicators, 1,262 laws and policies were identified and analyzed for their alignment with the international legal standards. Analysis shows significant gaps. 62% of all national laws are aligned. Nearly every country has multiple areas where national laws are not yet aligned with the Care Compact. Though alignment is feasible. In 5 of 6 regions at least one country has laws aligned on all indicators. Geographic region was not a significant predictor of alignment, while income level was only weakly associated. Comparing the key legal issues facing health and care workers, well over half of countries studied are fully aligned with the Care Compact on occupational health and safety, fair remuneration, enabling work environments, freedom of association, and collective bargaining. Approximately 50% of countries studied are fully aligned on protections against violence and harassment in the workplace and whistleblower protections. But less than 25% are fully aligned on access to health services in occupational settings and equal treatment and non-discrimination. Together this analysis highlights the need for, and opportunity of, law reform in countries throughout the world to elevate and protect the rights and well-being of health and care workers and, in doing so, improve health systems.

## Introduction

Health and care workers are the lifeblood of health systems, and their ability to function optimally is deeply dependent on their contexts. The world cannot fight pandemics or achieve

Protection and Rights.'" OSF. May 17 and Sept 13. doi:10.17605/OSF.IO/4XCBW.

**Funding:** The legal research, coding, and analysis for this article were funded by a grant from the World Health Organization (WHO, https://www.who.int) to Georgetown University (Grant number 70747). Dr. Matthew M Kavanagh served as PI. No other sources of funding contributed to the study. No authors received salary from commercial companies. Authors Giorgio Cometto, Catherine Kane, and James Campbell are employees of the World Health Organization and participated in study design, data collection and analysis, decision to publish, and preparation of the manuscript.

**Competing interests:** The authors have declared that no competing interests exist.

universal health coverage, let alone realize the right of everyone to the highest attainable standard of health, without these workers [1–4]. Yet health and care workforce challenges currently hinder progress toward realization of national and global health goals. The World Health Organization (WHO) projects an estimated global deficit of 10 million health workers by 2030, largely concentrated in low and middle income countries [5]. Population ageing, gender inequities and unmanaged international migration of workers, among other factors, can exacerbate pre-existing inequalities and deplete the workforce in countries already most affected by shortages [6].

The COVID-19 pandemic deepened both public and government recognition of the importance of health and care workers, and the importance of meeting their needs. 80,000 to 180,000 of these workers lost their lives to COVID-19 between January 2020 through mid-May 2021, based on data reported to WHO [7]. Health and care workers were at particularly high risk of infection, especially early in the pandemic, due to poor infection prevention and control measures, including a lack of access to sufficient personal protective equipment (PPE) and other occupational health and safety measures [8, 9]. Moreover, health and care workers often lacked access to tests, paid time off, sick leave, adequate treatment, and COVID-19 vaccines [10]. Investigators documented increased rates of mental health issues among health and care workers due to facing high rates of illness and death, extended working hours, exposure to higher levels of patient pain, suffering and death; and other stressors related to their own heightened exposure to the virus [11, 12].

In 2021, recognizing the unprecedented and multi-faceted challenges health and care workers faced during the pandemic [13], the 74th World Health Assembly (WHA) requested WHO to develop a Global health and care worker compact (Care Compact) to guide efforts to protect and safeguard health and care workers [14]. The Care Compact outlines the existing obligations and commitments under both binding and non-binding international law, and provides additional guidance to countries on how to best "protect health and care workers and safeguard their rights, and to promote and ensure decent work, safe and enabling practice environments free from racial and all other forms of discrimination" [15]. The following year, the WHA called on countries to use the Care Compact "to inform national review, action, and implementation to protect and support health and care workers" and asked the Director-General of the WHO to assist countries in protecting health and care workers and safeguarding their rights as defined in the Care Compact [16].

The Care Compact itself details the international legal framework that exists for supporting health and care workers and safeguarding their rights. It articulates the norms and standards that apply to four domains–preventing harm, providing support, ensuring inclusivity and safeguarding rights–comprised of ten issue areas including occupational health and safety, access to health services, protection against violence and harassment, equal treatment and non-discrimination, enabling work environments, and freedom of association. Further, it suggests management and policy actions. The Care Compact itself is not a legally binding instrument. Indeed, its guidance is couched in the language of recommendations [17]. However, much of the international legal framework that it is based on, such as human rights and labor law, is legally binding. Thus, the recommendations have a strong normative foundation originating in the legal framework that already existed before the Care Compact was developed.

Significant gaps in domestic law and policy exist, and implementation challenges remain [18]. There is a lack of research that analyzes national laws and policies that apply to health and care workers and that assesses the variation across countries.

Measuring, tracking, and comparing the laws and policies of countries are crucial tools for international governance. Comparison facilitates monitoring progress and compliance with human and labour rights obligations [19, 20]. Collecting and analyzing laws and policies

against internationally recognized standards and obligations improves transparency, facilitates study within and across regions, enables government and non-government stakeholders to have a solid understanding of the legal and policy environment, and empowers more informed advocacy for public policy change [21–23]. In this case, analyzing how national laws compare on indicators benchmarked against the Care Compact standards enables cross-country research to help stakeholders learn from similarly-situated states and consider adoption of legal and policy instruments, while also increasing the ability of civil society to hold their leaders accountable to international law [24]. When analyzed in tandem with epidemiological and other data, such as those about health and care worker retention and levels of violence against health and care workers, this information can contribute to a better understanding of how such laws and policies can contribute to health workforce policy and strategies as well as health outcomes.

The research objective of this article, therefore, is to measure alignment of national laws with the member state-recognized Global Compact to better understand the legal environment within which health and care workers live and work. This can both serve an information function, providing a baseline assessment of the current status, and support the implementation of the Care Compact. National laws, policies, or other measures (circulars, directives, guidelines, etc.) of 182 countries were measured. Each law and policy is evaluated against one of the ten issue areas of the Care Compact, and classified further by their alignment with underlying norms and standards. In light of the diversity of legal environments across each country, any law, policy, or measure that held sufficient force to ensure the protection of the norm or standard assessed was considered. We present the results of this exercise below, which provides a starting point from which law and policy reform can be considered to strengthen protection and rights of health and care workers and to improve both health care delivery and worker retention and attraction.

## Methodology

We present the empirical findings and analysis of national law and policy environments on health and care workers' protection and rights across 182 countries and comparison with the international standards defined by the Care Compact. We draw on methods from policy surveillance–the systematic, scientific collection and analysis of laws—and from comparative social science research [21, 22, 25]. As such, we began with conceptualizing indicators as proxies for each of the 10 issue areas within the Care compact; next, we defined a specific coding question and coding rule to enable law and policy analysis; and finally we conducted a search and analysis strategy designed to generate empirically valid measures of the legal environment [26, 27]. Below we describe specifically (A) the indicators developed and (B) the research strategy for collecting and coding legal texts and information to generate data on those indicators which are summarized in Fig 1. Using these methods, data was able to be generated for 182 of the 194 WHO member states, while for 12 states we were unable to generate data because neither the primary documents nor official reporting were able to be obtained. For our final analysis, we included countries with data for at least 3 of the 10 indicators.

## A. Defining and coding health and care worker law and policy environments

The Care Compact comprises of 10 issue areas across 4 domains derived from international labour and human rights legal instruments. It uses the WHO definition of health and care workers, which includes all those workers whose primary intent is to provide health and care services [28]. This covers both health professionals, such as doctors, nurses, associated health

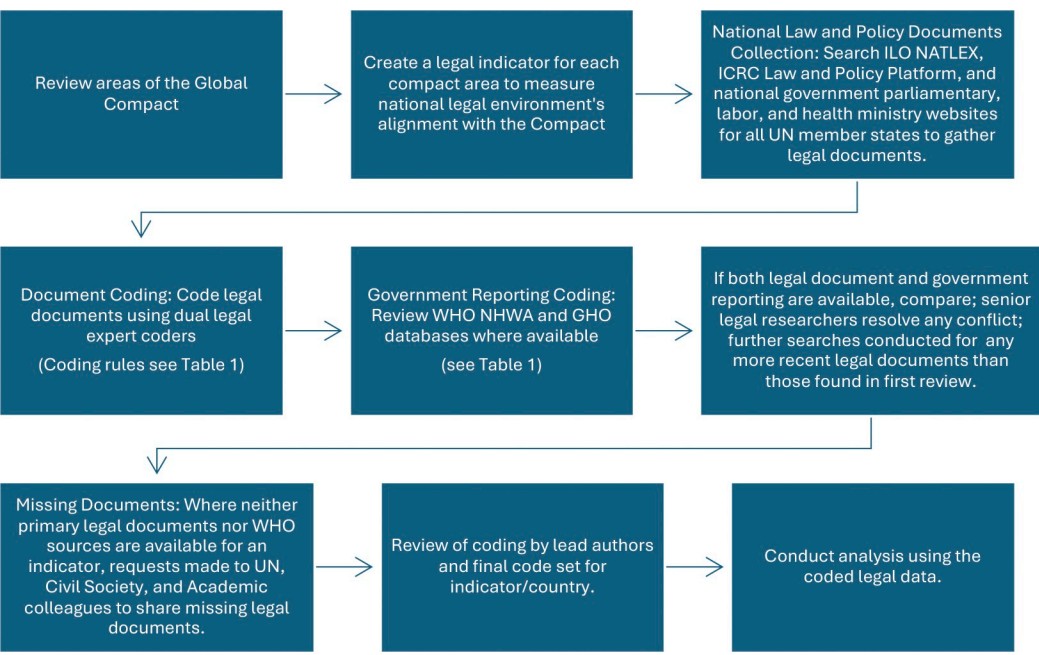

**Fig 1. Methodology for collecting and evaluating national laws.**

professionals, and community health workers, as well as care workers who provide direct personal care services at home, in health care, and residential care facilities. The latter include both health care assistants–workers who provide personal care at various institutions such as hospitals, clinics, and residential nursing facilities–and workers who provide care at homes, including routine assistance with activities of daily living.

Indicators were developed for each of the ten issue areas. Hundreds of laws, policies, and other measures fall within the scope of the Care Compact and relate to the rights of health and care workers—the indicators chosen are intended as tracers to represent this broader context. For each indicator, a coding question was developed to provide a specific, measurable indicator to research and enable cross-national analysis and data collection (See Table 1). Following literature in social science, indicators and codes were developed to be valid and sensitive to the underlying legal standards being measured, meaningful and easily interpretable, comparable across countries and contexts, and good markers of differences between national legal environments that can be consistently measured over time [26, 29].

The proposed indicator list was developed in consultation with, and reviewed by, WHO technical focal points as well as an external group of stakeholders composed of relevant UN organizations, occupational groups, labor unions and civil society organizations active in human resources for health through the International Year of Health and Care Workers Steering Committee (see S1 Text). Where possible, the indicators align with existing measurement instruments, in particular the National Health Workforce Accounts, to facilitate data collection and measurement from official sources and collection mechanisms (see S2 Text for details) [30].

A codebook was developed defining how to assess laws and policies using various source documents. The codebook was developed iteratively, informed by initial research conducted during a "scoping" stage, until a final version was established to assess alignment with the Care Compact.

**Table 1. Global health and care workers indicator questions, coding, and sources.**

| Global Compact Element | Indicator Question | Coding Rule (Aligned (Y), Partially aligned (P), and Not Aligned (N)) | Source(s) for Analysis |
|---|---|---|---|
| **Protection from occupational hazards** | Does the country have national policy instruments/laws for occupational health and safety that cover all health workers and require all health workers to have access to protective equipment or supplies? | Occupational hazard laws/policies that explicitly guarantee access to personal protective equipment to all health workers are aligned (Y) while those that cover all workers but do not require PPE access are coded partial (P) and those that do not cover all health workers are not aligned (N) regardless of PPE status. | • National law and policy documents<br>• WHO Global Health Observatory Database |
| **Providing health services for health and care workers** | Does the country have policies/laws that allow for a national package of health services for health workers that includes mental well-being? | Laws/policies that ensure all health workers are eligible to receive free medical coverage for occupational diseases or injuries, including mental health were classified as aligned (Y). Those with health coverage that did not include mental health conditions and well-being, were classified as partially aligned (P). Laws and policies that did not cover all health workers were classified as not aligned (N). | • National law and policy documents<br>• WHO National Health Workforce Accounts Data Portal (Indicator 4–01.7) |
| **Protection against violence and harassment** | Does the country have national policies/laws for the prevention of harassment and violence against all health workers? | National laws are classified as aligned (Y) if they include measures for the prevention of attacks against all health workers, which may include heightened penalties, required reporting, mandatory training, security measures, protection from immigration enforcement for reporting harassment, and the right to depart threatening situations; (N) indicates no such laws. | • National law and policy documents<br>• WHO National Health Workforce Accounts Data Portal (Indicator 4–01.6) |
| **Protection against attacks in situations of fragility, conflict, and violence** | Does the country have laws and/or policies that incorporate all of the Geneva Convention and Additional Protocol protections of health and humanitarian workers, health facilities, transport, and patients during armed conflict into national law? | Laws incorporating the full extent of the Geneva Convention protections of health workers and infrastructure into domestic policy or law are aligned (Y); those that cover only a subset are partial (P); and those lacking incorporation are not aligned (N). | • National law and policy documents |
| **Equal treatment and non-discrimination** | Does the country have a law that prohibits discrimination in the workplace, on internationally protected grounds, that includes all health and care workers? | Laws/policies that prohibit workplace discrimination on all internationally recognized grounds are aligned (Y); on only some grounds are partial (P); and covering no grounds or not covering all health workers are not aligned (N). | • National law and policy documents |
| **Fair and equitable remuneration** | Does the country have national policies/laws covering health workers that: 1) regulate working hours and conditions, and (2) regulate a minimum wage? | National laws that cover both minimum wage and working hours/conditions for health workers are aligned (Y); one but not the other is partial (P); and neither is not aligned (N). Hours and conditions are coded based on availability of basic protections for at least some health workers while the minimum wage sub-indicator requires at least some health workers to receive a minimum wage under national law or policy. The focus is on the regulation of a minimum wage, and not the level. | • National law and policy documents<br>• WHO National Health Workforce Accounts Data Portal (Indicator 4–01.1 & 4–01.2) |
| **Social protection** | Does the country have national social protection policies/laws that, at minimum, provide parental leave to all health workers? | Social protection including parental leave dictates alignment (Y/N).<br>Laws that include parental leave, but cover some but not all health workers, are partial (P). For example, some countries have different social protection measures for public and private sector employees. | • National law and policy documents<br>• ILO Global Care Policy Portal<br>• WHO National Health Workforce Accounts Data Portal (Indicator 4–01.3) |
| **Enabling work environments** | Does the country have policies/laws for a national system of professional development for health workers that includes: 1) Access to continuing professional development, and 2) In-service training for the health workforce? | Laws that cover both continuing and in-service training are aligned (Y), those covering only one are partial (P), and those with neither are not aligned (N). | • National law and policy documents<br>• WHO National Health Workforce Accounts Data Portal (Indicator 2–07.5) |

*(Continued)*

**Table 1.** (Continued)

| Global Compact Element | Indicator Question | Coding Rule (Aligned (Y), Partially aligned (P), and Not Aligned (N)) | Source(s) for Analysis |
|---|---|---|---|
| **Freedom of association and collective bargaining** | Does the country have policies/laws that protect the right to join an independent union or similar workers' organization that apply to some health workers? | National laws that protect health workers' right to join an independent union are aligned (Y). Independence is assessed per international labor norms on union registration, formation, and membership independence from government [54]. Laws without true independence are classified as partial (P). No protections mean not aligned (N). | • National law and policy documents |
| **Whistleblower protections and freedom from retaliation** | Does the country have policies/laws providing whistleblower protections that include freedom from retaliation and guaranteed confidentiality to all health workers? | National laws that provide whistleblower protections to all health workers and include freedom from retaliation and guaranteed confidentiality are aligned (Y). Those that covered one but not both are partial (P) and those that covered neither are not aligned (N). | • National law and policy documents |

For all indicators we developed a coding that differentiates between "aligned" (or yes on our question) and "not aligned" (no)—specifics of which are explained below for each area. In most, but not all, we were able to distinguish a meaningful "partial" indicator which should be understood as at approximately half way toward alignment. While in some cases sub-national analysis would be appropriate it was not feasible across all countries nor would sub-national analysis be comparable, so we limited the analysis to the national level in scope and recognize this as a limitation.

**1. Ensure occupational health.** Without adequate health and safety measures, health and care workers themselves are at higher risk of disease, injury, and death, not only violating their rights but also preventing their patients from receiving safe care. A dearth of occupational health and safety reduces health and care worker retention. As the shortage of high-quality masks and other protective gear during the COVID-19 pandemic spotlighted, the lack of personal protective equipment heightens health and care workers' risk of infection [9]. Furthermore, international labor law expressly requires employers to provide employees with protective gear, as do UN General Assembly and WHA resolutions. [31–33]. These existing international standards led to the selection of protective equipment or supplies as a relevant and measurable tracer to assess the legal environment on occupational health. For the indicator of preventing harm from occupational hazards, national laws were classified based on whether they 1) protect all health workers and 2) ensure access to protective equipment or supplies.

**2. Provide health services.** Providing adequate health care to health and care workers is critical for maintaining a strong workforce. The Care Compact states the importance of providing comprehensive occupational health and health care services for all health and care workers that meet their "physical, mental, and psychosocial needs" [17]. To adequately assess the differences in law and policy environments across all countries, the indicator selected measures the coverage of all health workers and the inclusion of mental health services as a proxy for a comprehensive package of health services. National laws were thus classified based on whether they have national policies or laws that 1) provide health services for all health workers, particularly medical services for occupational diseases or injuries, and 2) include mental well-being or psychological support.

**3. Protection against violence and harassment.** To ensure health and care workers can provide critical and important care, they must be protected from violence, discrimination, and exploitation of any kind. Protection against violence and harassment is a part of both

international human rights law and labor law, yet health and care workers face particularly high levels of violence [34]. Laws and policies protecting health workers in particular are needed given their needs and vulnerabilities, which are different from the general public, and must be in line with the obligations and commitments in human rights law, labor treaties, resolutions, and declarations, and other international instruments [35]. There is no single law or policy that offers a clear solution, but rather a suite of measures that States may take to prevent this violence [36]. Therefore, the indicator selected encompasses a wide variety of possible measures that States can take to protect health workers against violence and harassment, coming at the problem from multiple angles including deterrence, training and security, and health workers' right to keep themselves safe. Countries with national law or policy that includes at least one of the following measures: (a) Heightened penalties for violence against all health workers; (b) Requirements on reporting violence against all health workers; (c) Mandatory training on workplace harassment; (d) Health workplace security measures; (e) Explicit protections for care workers against any immigration enforcement consequences if they report workplace violence; and/or (f) The right of all health workers to remove themselves from situations where they reasonably believe there is a serious and imminent risk of violence or harassment against them.

**4. Protection against attacks in situations of war/fragility, conflict, and violence.**
Attacks on health care during armed conflicts and other violence have been on the rise, with the number of health workers killed exceeding 230 in 2022 during the nearly 2,000 attacks on health that year [37]. This is even as attacks on health represent serious human rights violations. International humanitarian law requires states to ensure the protection of medical and humanitarian personnel, as well as their means of transport, equipment, and working environments. Indeed, the foundational treaties of international humanitarian law, the Geneva Conventions and their Additional Protocols, have been universally or widely ratified (respectively), and underpin customary international law that is binding on all states [38, 39]. Given the centrality of the Geneva Conventions and their Additional Protocols to international humanitarian law, and their wide-ranging protections of health workers and their infrastructure, this indicator examines whether these instruments' protections have been incorporated into national law. Specifically, national laws were classified on the state of the domestic incorporation of all or some of the protections established by the Geneva Conventions and their Additional Protocols for health and humanitarian workers, health facilities, transport, and patients during armed conflict. National laws found to incorporate only a subset of the Geneva Convention protections that relate to health work in conflict settings were classified as partially aligned. For example, some national laws specifically criminalize the killing of a protected person as defined by the Geneva Conventions, but do not include acts of violence or attacks against such protected persons as a part of the crime.

**5. Ensuring equal treatment and non-discrimination.** Guarantees against discrimination based on people's identity are foundational to international human rights law [40]. Discriminating against health workers based on their identity, like their gender, race or religion, causes individuals harm and can in turn, harm worker retention, including among marginalized health and care workers who may be particularly well positioned to reach those poorly served by the health system. Women comprise 67% of the health and care workforce yet earn 24% less than their peers who are male, and they are underrepresented in leadership positions [41]. Our indicator therefore assesses whether countries' laws prohibit workplace discrimination against all statuses against which discrimination is prohibited under international human rights law. National laws were assessed on whether they have specific laws that prohibit discrimination in the workplace, with the requirement that they 1) cover all health and care workers on 2) all internationally protected grounds, as defined by the Committee on Economic,

Social and Cultural Rights in its General Comment No. 20 on Non-Discrimination [42]. To assess this indicator, laws and policies beyond a simple constitutional equal protection provision were researched due to the importance of a legal framework to implement such protections in the workplace, such as by providing redress, specifying what workplace discrimination covers (e.g., in hiring or pay), establishing penalties for violations, and defining what discrimination entails (e.g., if intent is required).

**6. Providing fair and equitable remuneration.** Ensuring fair and equitable remuneration for all health and care workers is essential to provide these workers with safe, healthy, supportive, and dignified conditions of work [43]. Fair payment is particularly important for addressing gender inequalities [44]. Country alignment was determined based on whether national laws or policies covering all formal health workers both 1) regulate working hours and conditions through measures such as overtime protection, maximum working days, or guaranteed rest days; and 2) also establish a national minimum wage. These were identified as appropriate indicators for a regulatory environment in line with the Care Compact standards due to their importance under international human rights law [45]. Given the complexity and diversity of how countries regulate fair and equitable remuneration, a multi-part analysis was utilized.

We note that this indicator covers only *formal* health workers, and thus does not cover informal workers or volunteers—even though many of those "volunteers" are working many hours, even full-time in some cases. It is important that all health and care workers, including community health workers, receive a decent wage commensurate with hours worked, tasks and roles undertaken, responsibilities, qualifications and risk, with consideration to cost of living [46–49].

**7. Social protection.** Health and care workers often lack social protections like unemployment insurance, child-care, and family leave. Health and care workers who are not paid for providing services also lack access to other social benefits [50]. To adequately distinguish countries based on their law and policy environments for social protection, we focused on one protection measure as a tracer. Parental leave–that is leave available to either parent, regardless of gender, allowing them to care for an infant or young child–was selected based on existing norms and obligations in international labor law, consultations with stakeholders, and current global trends for paid leave practices. We also chose it to align with WHO's National Health Workforce Accounts. We recognize that this is a narrow indicator in light of the expansiveness of social protections [51, 52].

**8. Enabling work environments.** Supportive supervision, effective management, and opportunities for feedback create stronger and safer conditions for health and care workers. Enabling work environments also are linked to reducing gender disparities in health and care work. As defined by the Care Compact, this includes access to health information through professional development and lifelong learning and ensuring that the health and care workers can meet personal and family responsibilities [53]. A multi-part assessment focusing on access to health information for job performance was selected as a tracer for this indicator. National laws and policies were assessed on whether the measures required health and care workers to have access to 1) continuing professional development opportunities, making training beyond routine clinical updates available that could include supporting a wide range of competencies [30]; and 2) in-service training to maintain and expand health care skills and competencies.

**9. Guaranteeing freedom of association and collective bargaining.** The right to join a union and enable workers to collectively bargain to ensure their rights are respected and needs are met is a human right in and of itself, as well as a core labor right [40, 54]. Effective unions can protect and advance the rights of health and care workers, encompassing the full range of the Care Compact areas. To fully realize this right, health and care workers must have the ability to form and operate trade unions of their choice, free of government control [55]. Without

independence, unions might not robustly represent the interests of workers and face conflicting government interests–such as reducing spending, avoiding unrest, and limiting the government's own accountability. As such, this indicator assesses whether a country has national laws or policies that protect the right of health workers to join an independent union or similar workers' organization, beyond a constitutional provision on freedom of association.

**10. Whistleblower protections.**   The right to whistle blowing is inherent in international human rights and labor law, which includes the right to impart information, and implicates the rights to health and to just and favorable working conditions, among others [40]. For example, the ILO Violence and Harassment Convention (2019), and the UN Convention Against Corruption (2004) include whistleblower protection. These protections are critical for protecting the health and safety of individuals and the community, supporting workers to inform authorities about unsafe conditions for health workers or patients or informing the public that the government is hiding a public health threat. Prohibiting retaliation from employers and guaranteeing confidentiality are two core practices that can support health and care workers who help expose malfeasance [56]. As such, this indicator assesses whether national law and policy include both protections from retaliation and guaranteed confidentiality for all health workers who report waste, fraud, abuse, corruption, or dangers to public health and safety.

Table 1 summarizes the indicator questions and coding rules developed to translate information about law and policies into a measure of alignment with the Care Compact.

## B. Methodology for collecting and evaluating laws

Guided by the content of the Care Compact and our specific indicator questions, we searched for primary legal documents in the form of legislation, regulations, national policy and strategy documents, circulars, and directives to assess countries' legal environments.

The research strategy focused on collecting primary source documents ("National law and policy documents" in Table 1), making use of several online document repositories, such as the International Labor Organization's NATLEX database and the International Committee of the Red Cross Law and Policy Platform, as well as national government and parliamentary websites including those of ministries of health and labor for all UN member states. Where possible, we conducted searches in the national language of the target country and we used a multi-lingual team of researchers fluent in many languages for data collection and coding, with default to translation services where needed. Where legal texts were not available from these searches, we used networks of civil society, government, and intergovernmental organization officials to gather further documents. In some instances, we conducted a selective search of secondary literature for areas or analyses that required additional information or verification from the initial research, such as regarding the independence of union formation or domestic incorporation of Geneva Convention protections. These included research papers, domestic news outlets, publications of and consultations with civil society organizations and national experts on health and care worker rights, and reports from verifiable international organizations or non-government organizations. We also reviewed some of these materials at the outset of the analysis of a specific country to better understand the countries' legal environment and identify the names of relevant laws and policy documents to source. We have created a repository of all the primary source legal documents used, which can be found at https://www.hcwpolicylab.org/.

In addition, as a distinct primary source we also used official government reporting through international organizations for indicators with available data, in particular the NHWA and other components of the Global Health Observatory Database (GHO) and the International

Labor Organization Global Care Policy Portal [57–59]. These government reports were considered primary sources since they are validated government reporting. Where more than one source (e.g. legal document and official WHO reporting) was found for a country, any conflicts were investigated and adjudicated by principal researchers. Table 1 indicates sources used for each indicator and S2 Text has more information specific sources.

The laws in each country were assigned as aligned (Y), partially aligned (P), or not aligned (N) based on the coding rules described in Table 1. Legal documents were coded to assess and measure the legal and policy environments using comparative legal interpretation and content analysis methods [60–62]. We used a dual-coder strategy with checks for intercoder reliability and final coding review by senior researchers. Following the practice of policy surveillance, the focus of the analysis was on the content of the law or policy in the country, rather than on the extent to which the measure has been implemented [22]. The full codebook can be found in S4 Text.

Countries where research did not yield any law or policy answers were classified as no data —reflecting that we cannot adjudicate whether there is no law or whether such a measure exists but could not be found.

We conducted a series of descriptive analyses using this data, which primarily consists of arithmetic counts and averages (means) that are displayed in the results section below. To compare and analyze significant differences in mean indicator alignment between income levels and WHO regions, a one-way analysis of variance (ANOVA) was conducted. This statistical method was used to test whether income levels and WHO regions were significant predictors of alignment values. Post-hoc pairwise comparisons were performed using the Tukey-Kramer method to analyze specific differences between regions and income levels. This approach allowed for a detailed analysis of alignment differences while addressing the potential for multiple comparisons. Data and analysis results are available in S3 Text.

## Results

We collected empirical legal environment information from 182 countries across all 6 WHO regions and assessed the extent to which they aligned with international norms and obligations on health and care workers' issues, as described in the Care Compact. In the graphs that follow, laws are categorized as aligned (labeled 'Y' for a yes answer to the coding question), partially aligned (labeled 'P') where relevant, and non-aligned (labeled 'N' for a no on the coding question) based on the coding rules in Table 1. For some indicators, no relevant law or policy could be found for some of the countries analyzed or a final determination was not possible. In those cases, we exclude them from our analysis.

### Global alignment with international norms on health & care worker protection & rights

One way to assess the world's current overall alignment between national laws and international norms can be seen by looking at what portion of all the world's national laws are aligned with the international norms in the Global Compact across all countries and indicators. While many countries have worked to ensure their laws and policies that govern the health and care workforce align with international human rights and labor norms, the data we collected show notable gaps. Across all countries and indicators we were able to analyze 1,262 national laws and policies. As shown in Fig 2 below, of all the national laws analyzed 62% are fully aligned, 16% are marked as partially aligned, and another 22% are categorized as non-aligned.

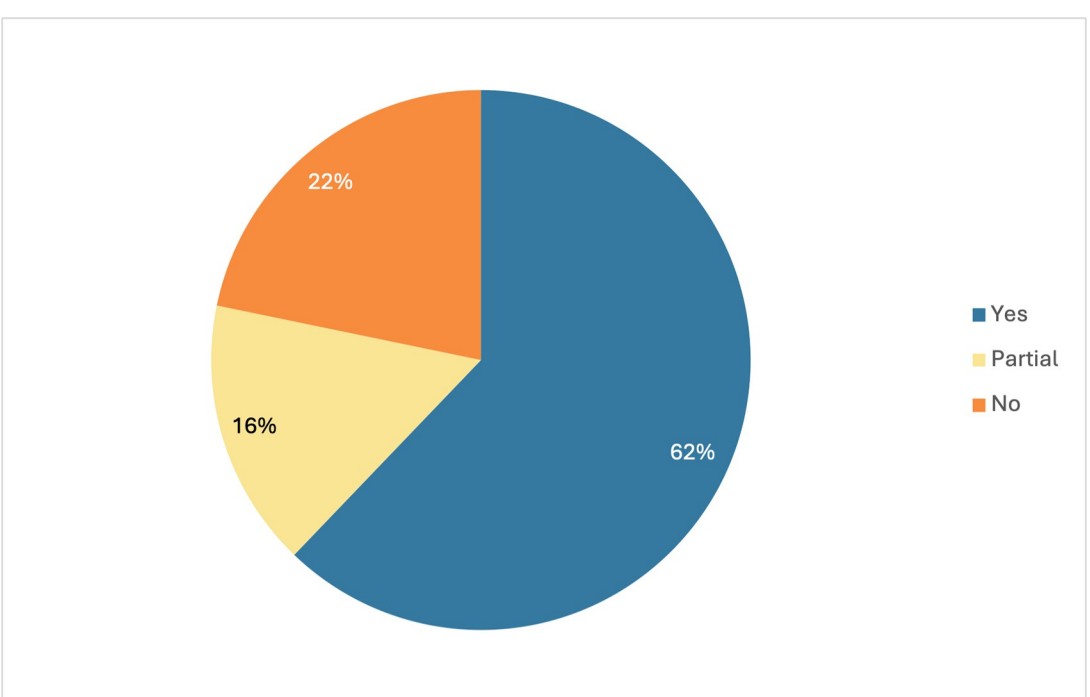

**Fig 2. Global alignment of national laws with international norms on health and care workers.**

### Variation across legal areas

There is substantial variation between legal areas on the relative alignment with international norms. For none of the indicators for the Global Health and Care Worker Compact are all countries fully aligned. In contrast, for every indicator there are countries where national laws are aligned with the Care Compact. Fig 3 shows the relative alignment of national laws and policy environments globally for each of the 10 areas of law and policy explored. Since we were not able to find laws and policies for every country on every indicator, we included the N of countries for which we have data out of the total researched (182). Data informing the graphs in Fig 3 are included in S3 Text. Each of the statistics provided in the text below is based on a denominator of the countries with available data.

The most aligned area globally is remuneration and fair employment, where 93% of countries (160/172 countries for which we have data) have both a minimum wage and regulation of working hours for all formally recognized health workers. 6% of countries are partially aligned, regulating either minimum wage or working hours but not both. Just 1% of countries are not aligned. However, we note that this covers only *formal* health workers.

On occupational health, a majority of countries (70%, 107/153) are aligned with the Care Compact, having national laws and policies for occupational health and safety that cover all health workers and require them to have access to protective equipment or supplies. 18% of countries have occupational health laws that cover all health workers but do not explicitly provide access to protective equipment. 12% of countries lack occupational health laws that cover health workers.

Health services for health worker policies are not as aligned. While three quarters (69%, 94/137) of countries researched have some form of health services for all health workers, only 20% of these include services for mental health conditions and well-being. 12% of countries do not provide health service packages that include all health workers.

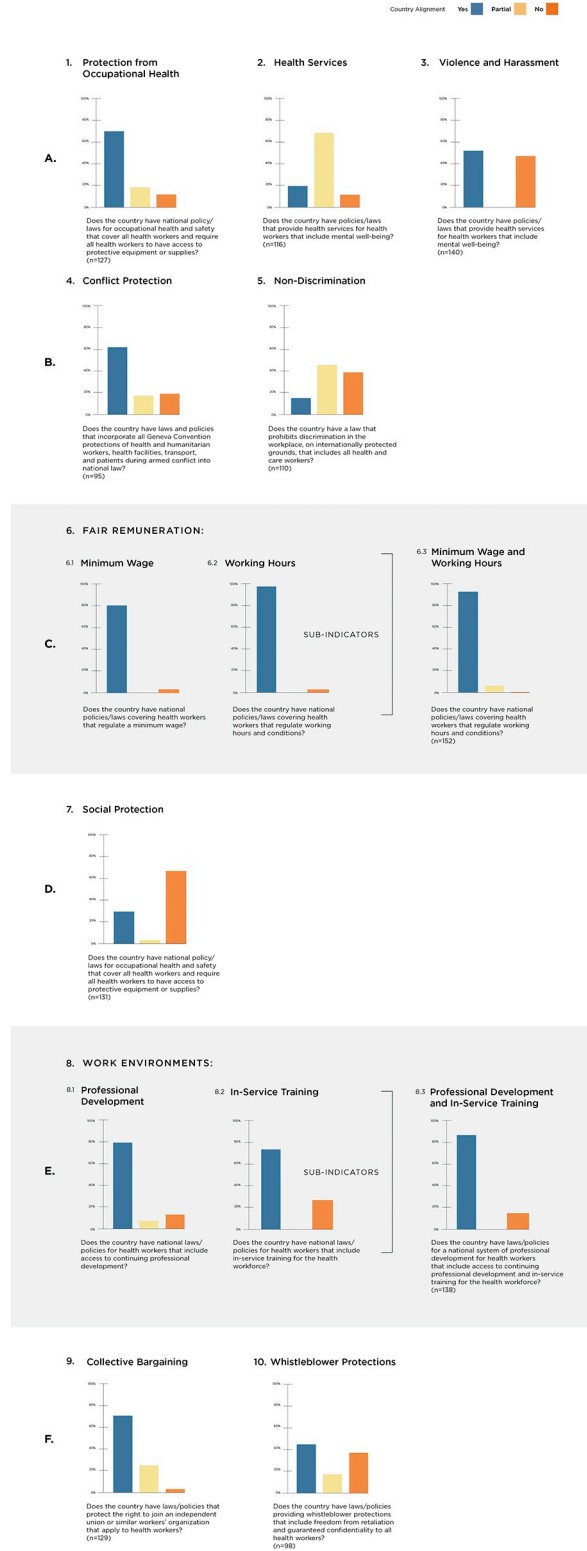

**Fig 3. Global legal alignment across areas of health & care worker protection and rights.**

More than half of countries researched have (53% 86/163) national laws and policies specifically aimed aim to prevent violence and harassment of all health workers.

Looking to contexts of fragility, conflict, and violence, 63% (71/113) of countries have national laws and policies that incorporate all Geneva Convention protections of health and humanitarian workers, health facilities, transport, and patients during armed conflict. 18% have legal stipulations, but do not incorporate the full extent of the Geneva Convention protections. 19% have no legal stipulations that incorporate any of the Geneva Convention protections.

The least aligned legal area globally is addressing equal treatment and non-discrimination, with only 16% (20/128) of law and policy environments aligned with global norms and prohibiting workplace discrimination on all internationally recognized grounds. 44% prohibit discrimination on some but not all protected grounds, and 40% are coded as lacking non-discrimination workplace protections for all health and care workers.

Only 30% (46/153) of countries have national social protection policies and laws that, at minimum, provide parental leave to all health workers. More than half (67%) of countries do not provide this level of social protection to health workers at all. A small portion of countries (3%) studied have national social protection policies and laws that provide parental leave to some but are only partly aligned as the parental leave measures do not cover all health workers.

Enabling work environments in about 79% (138/175) of countries fully meet the Care Compact standard indicator and actively include continuing professional development and in-service training to help ensure all health workers can do their job effectively. A relatively small portion (8%) provided access to one, but not both. 13% of countries researched are not aligned in this area.

Freedom of association and collective bargaining is relatively well aligned, with 72% (111/155) of countries have laws and policies that protect the right to join an independent union or similar workers' organization. Among 26% of countries, that right is not truly independent. 2% have no law or policy enabling health workers to form or join a union.

Fewer countries have strong whistleblower protections, with just under half (45%, 52/115) of national laws ensuring both freedom from retaliation and guaranteed confidentiality. 17.5% of countries researched have laws or policies that provide whistleblower protections, but only include one of the two important protections while 37% have no whistleblower protections.

Fair remuneration is the indicator for which we were able to find widest coverage of laws, in significant part due to the ILO collection of laws, while conflict protection and whistleblower protection which have received less political attention has the lowest coverage, thus we have the lowest confidence in our alignment Figs for the latter. This reflects the important global agenda-setting role [63] for international organizations reflects relatively low priority for whistleblower protections in global health and labor.

## National variation

Major variation exists across countries in the world. The average alignment of countries' laws with the Care Compact is 63% of indicators studied. But as shown in Fig 4, some countries' national laws are highly aligned with the Care Compact, such as Spain, Namibia, Brazil, Bhutan, and Canada with some of the highest portion of fully "aligned" laws. In 5 of the 6 WHO regions, all except South-East Asian Region (SEAR), there are countries where the laws are aligned on all indicators for which we have data. Countries least aligned (less than 20%) include countries from the same 5 of the 6 WHO regions such as Belize, Bosnia and Herzegovina, Laos, and the Bahamas. Fig 4 shows the alignment for all countries at a granular level across indicators.

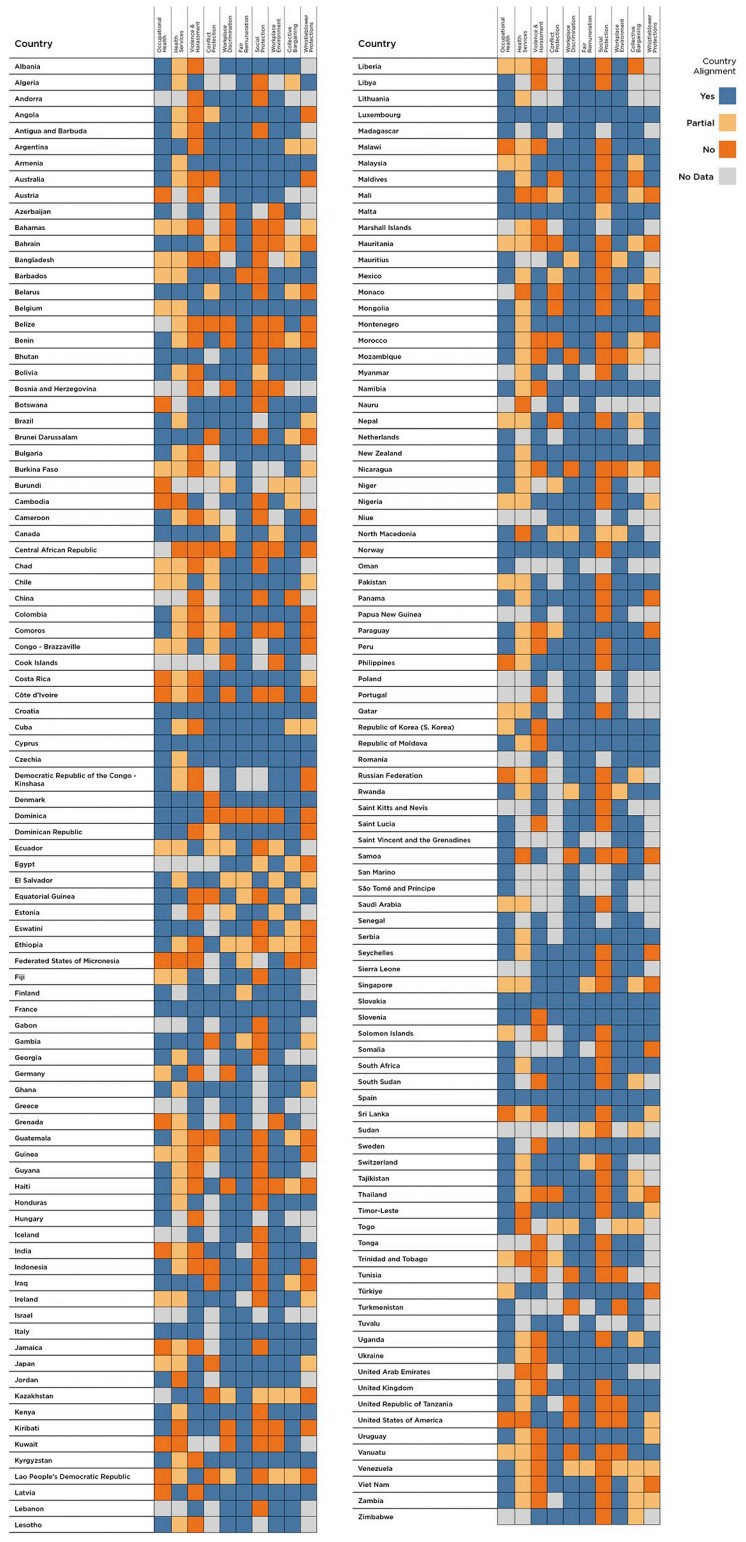

**Fig 4. Alignment across all 10 indicators for 182 countries researched.**

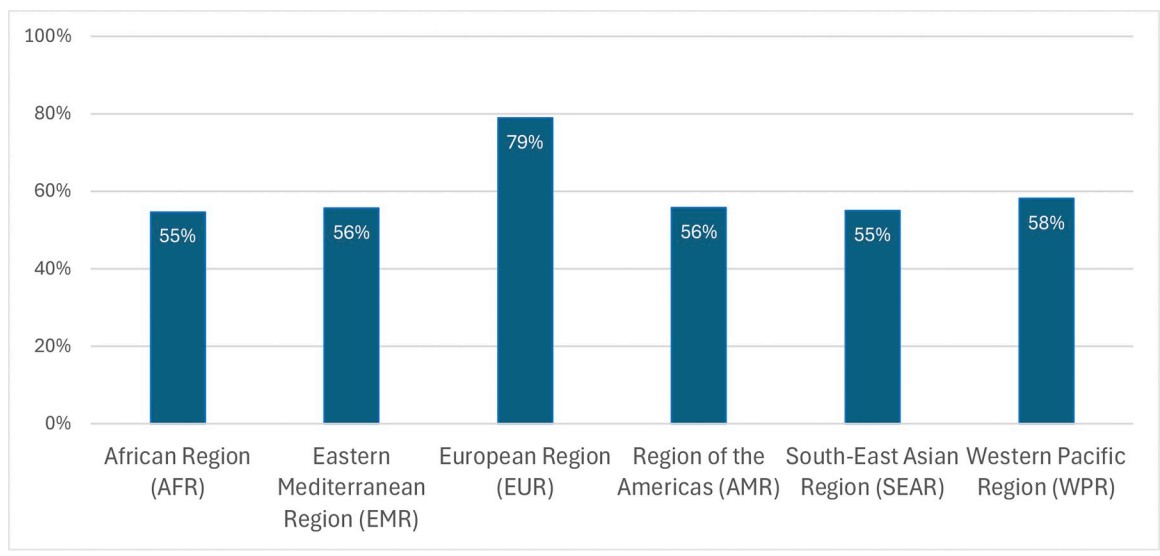

**Fig 5. Percentage of national laws that are "aligned" and international norms on health and care workers by region.**

## Regional and economic variation

A region-wise analysis of alignment of national laws and policies with the international norms in the Global Compact shows alignment percentages between regions are close, ranging between 55% and 58%, with the European Region (EUR) being an outlier at 79% (Fig 5).

Overall, the trends we find on alignment span geography and income level. Indeed, there are countries at every income level and throughout the world that have both high alignment rates and low alignment rates. There is some support in the literature for the view that higher income countries are more capable of protecting rights and therefore more likely to have robust legal environments [64–66]. We find some support for that in the data, with high income countries aligned on 74% while low income countries are aligned on 47% and middle-income countries fell between (LMIC 54%, UMICs 63%). Income category was a significant predictor of alignment (p<0.01). However, these differences are not nearly as substantive as the differences between otherwise similar countries including neighbors. Following literature that shows that countries often transfer policies between neighbors [67, 68], we also explored regional differences. We find that overall region was a statistically significant predicator of alignment in the dataset (p <0.01), with the European region being the only significant outlier with a higher rate of alignment (79% aligned). Data and analysis supporting this regional and economic variation analysis can be found in S3 Text.

## Discussion

Overall, our analysis demonstrates that while many countries have established law and policy environments that offer protection and rights to health and care workers, there is significantly more that needs to be done in most countries to ensure these meet the global standards recommended in the Care Compact. More than a third of laws are not aligned with the Care Compact in the average country and many have adopted aligned laws and policies at much lower levels. But that does not indicate that these global norms are impossible to adopt. Considerable global progress for all Compact areas is possible, yet will require reform to ensure that the rights health and care workers hold are fully enshrined in national legislation and policies.

Notably, 105 countries are aligned with half or more of the indicators studied. This illustrates situations where national law and policy aligns with international law and standards to a significant degree, but some considerable gaps remain. Encouragingly, laws in many countries are aligned with a significant number of the indicators. Well over half of countries studied are fully aligned with the indicators assessing laws and policies on occupational health and safety, violence and harassment, conflict protection, fair remuneration, enabling work environments and freedom of association and collective bargaining. On the indicator assessing whistleblower protections, approximately 50% of countries studied are fully aligned. Results were very different for other indicators, though. Less than 25% of the countries studied are fully aligned with the indicators on access to health services in occupational settings and equal treatment and non-discrimination.

Written law and enforcement/implementation reflect different aspects of the empirical legal context. Written law by itself has important social, political, and behavioral impact that shapes public health outcomes [69]. From human rights and constitutions to laws on clinical medicine, social protection, drug criminalization, and beyond, researchers have shown that even before considering enforcement or implementation, the existence of legal texts structures daily life by directing how people understand they are to behave and how resources are distributed [70–74]. Our research falls into this area of focus—on the law *de jure*. At the same time, the existence of a positive law or policy is only the first step to fulfilling the rights of health and care workers. Protections without resources to enact them, laws without mechanisms to operationalize and monitor them, and rights without the capacity to enforce them have far less impact. In this case, for example, a mandate for ensuring all health workers have protective equipment or social protection program that goes unfunded and undeveloped will not have the intended effect. As such, particularly in environments where a significant number of Compact principles are enshrined, further study on implementation, execution, and financing would be particularly important. Even more importantly, work to ensure the implementation of laws matters. Our data can help differentiate between contexts where the law is insufficient compared to those where the law is aligned with the Care Compact but the lack of impact may be due to non-implementation.

Several next steps could enhance the value of this analysis. The research provides a data set to enable cross-sectoral analysis with epidemiological and health care workforce data, such as worker retention rates and distribution patterns, to reveal how laws and policies correlate with desirable national health outcomes. Future research should focus on adding more indicators relevant to the rights and well-being of health and care workers and filling the gaps for countries with missing data across all current indicators. Another next step would be to engage domestic stakeholders, including governments, unions, civil society and health and care workers themselves to better understand the implementation of national law and policy, and to drive policy reform, as well as implementation, regulation and enforcement, where needed.

Further investment in realizing the potential and promise of the Care Compact could entail developing more indicators across the Care Compact's 10 elements to capture all their aspects more fully. Given the level of marginalization that care workers face, especially home-based care workers, expanding indicators that currently focus on health workers to include explicit disaggregation by occupations (and including care workers) should be a particular priority.

## Limitations

This study makes use of legal and policy indicators that are necessarily reductive in nature and cannot capture the full complexity of the legal environment. Others have proposed the use of "legal environment assessments" that go deeper into specific legal areas and are aimed less at

finding comparative legal information than building qualitative analysis from which reform can be considered [75–77]. We support this strategy and are currently developing a Compact-specific assessment tool to enable specific legal analysis, and also help address gaps in our study to date. This study is also limited by the missing data. Having mobilized the existing resources available to us we have been able to marshal empirical legal information for 182 countries, most of which have information available on most of the Care Compact indicators. But there remain gaps. Wide coverage across geographic region and income level suggests only limited bias to the data along those axes, but some of the smallest states and those least connected to the international system are missing, which means this study does not well represent these countries. We do not think these are likely to dramatically change our findings or conclusions, but they are notable and further work to close them is ongoing. Specifically, the reliance of this study on laws and policies available from a review of databases, online sources, outreach to the authors' networks, and data collected by international organizations is a limitation of this study. While we did our best to make up for this limitation through pro-active outreach and search, that countries less engaged with the international community due to language, political context, or capacity are more likely to be missing from this analysis. We were unable within the scope of this study to look at the sub-national level, even though in some cases this would be appropriate in, for example, federalized systems. We determined this was not feasible across the number of countries and that doing so would make cross-national analysis impossible, but we recognize this as a limitation on the validity of our measures. In addition, this paper does not address the relative implementation of the laws and policies—a question that is outside the scope of this paper. We contend nonetheless, following strong evidence from legal epidemiology, that law in and of itself shapes human health and health systems and therefore is a critical factor to measure [27, 73, 78, 79]. This initial analysis also covered primarily health workers, with further research needed to understand the protection of care workers.

## Conclusion

Documentation of laws and policies on health and care workers offers new insights into the environment for health care workforce in countries around the world. Benchmarked against the Global Health and Care Worker Compact, this study provides a critical first global window into the incorporation of the human and labor rights of health and care workers into domestic law.

Most importantly, this analysis reveals major gaps in the national legal environment for the rights and protection of health workers. Notably, across the all the national laws reviewed, over a fourth were not aligned at even the minimal level captured by our indicators. This suggests a clarion call to action for governments around the world to use law and policy reform to improve the working conditions and, by reflection, the motivation, retention and efficacy of their health workforce. This has, to date, not Figs prominently on the international agenda.

The Global Health and Care Worker Compact—developed at the request of and adopted by the World Health Assembly—can offer important support for government officials. Research shows that most governments comply with most international commitments and that reporting can be an important tool for supporting implementation of international agreements—particularly "soft" law like the Global Compact [80]. Elevating and protect the rights and well-being of health and care workers worldwide is gaining in priority.

This article offers a baseline assessment that could serve as a basis for a coordinated law-reform agenda, providing a roadmap for action for lawmakers, governments, health and care worker representatives, unions, civil society and technical agencies. This article shows that in

every area of the Care Compact, there are governments that have aligned their national laws with its content—meaning that in almost every region and at every income level, governments will find peers who can share their experience and example laws. Beyond peer examples and sample legal text, barriers to law reform are likely to include a variety of considerations including the resources needed to expand protections, technical work on specific reforms, and competition for attention in the national political agenda. Each of these would benefit from a coordinated effort to turn the Care Compact from words into action. Improving the wellbeing of health workers and their ability to claim their rights is a good in itself, but it is also crucial to improving health systems in meaningful ways in the years ahead.

## Supporting information

**S1 Text.**
(DOCX)

**S2 Text.**
(DOCX)

**S3 Text.**
(DOCX)

**S4 Text.**
(DOCX)

## Author Contributions

**Conceptualization:** Matthew M. Kavanagh, Vishakh Unnikrishnan, Giorgio Cometto, Catherine Kane, Eric A. Friedman, Varsha Srivatsan, James Campbell.

**Data curation:** Matthew M. Kavanagh, Adi Radakrishnan, Vishakh Unnikrishnan, Eric A. Friedman, Varsha Srivatsan, Luis Gil Abinader.

**Formal analysis:** Vishakh Unnikrishnan.

**Funding acquisition:** Giorgio Cometto, Catherine Kane, Eric A. Friedman, James Campbell.

**Investigation:** Matthew M. Kavanagh, Adi Radakrishnan, Eric A. Friedman, Varsha Srivatsan.

**Methodology:** Matthew M. Kavanagh, Adi Radakrishnan, Catherine Kane, Eric A. Friedman, Varsha Srivatsan.

**Project administration:** Matthew M. Kavanagh, Giorgio Cometto, Varsha Srivatsan.

**Resources:** Matthew M. Kavanagh.

**Supervision:** Matthew M. Kavanagh, Giorgio Cometto, Catherine Kane, Varsha Srivatsan, James Campbell.

**Validation:** Giorgio Cometto, Eric A. Friedman, Varsha Srivatsan, Luis Gil Abinader.

**Visualization:** Vishakh Unnikrishnan.

**Writing – original draft:** Matthew M. Kavanagh, Adi Radakrishnan, Vishakh Unnikrishnan.

**Writing – review & editing:** Matthew M. Kavanagh, Vishakh Unnikrishnan, Giorgio Cometto, Catherine Kane, Eric A. Friedman, Varsha Srivatsan, James Campbell.

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
