## [Decision Letter · Decision Letter 0]

3 Jul 2024

PGPH-D-24-01084

Laws for Health and Care Worker Protection and Rights: A study of 178 countries

Dear Dr. Kavanagh,

Thank you for submitting your manuscript to PLOS Global Public Health. After careful consideration, we feel that it has merit but does not fully meet PLOS Global Public Health’s publication criteria as it currently stands. Therefore, we invite you to submit a revised version of the manuscript that addresses the points raised during the review process.

We look forward to receiving your revised manuscript.

Kind regards,

Veena Sriram

Academic Editor

Journal Requirements:

1. Please provide separate figure files in .tif or .eps format only and remove any figures embedded in your manuscript file. Please also ensure all files are under our size limit of 10MB.

Additional Editor Comments (if provided):

Thank you for submitting this excellent and important paper on the legal environment pertaining to health and care workers globally. The reviewers have noted the significant contributions of this work in their reports, and have also acknowledged the extensive efforts involved in undertaking this exercise. Please see their detailed feedback in their reports – my additional comments below. Thank you once again for the opportunity to engage with this important work, and look forward to the revised version.

1) Transparency in denominators for indicators – the authors note that “Each of the statistics provided in the text below is based on a denominator of the countries with available data.”. As discussed by both reviewers, it would be extremely important to acknowledge missing data by providing the denominator for each indicator in the text and in Figure 3.

2) Contributions – Further acknowledgement of the multilingual analyst team that conducted the analysis would be important, particularly given the diversity of languages present in the dataset. Reviewer 1 suggests a collaborator author group listing as part of the authors – another option would be a clearer indication of their role in the section on contributors. We would be happy to discuss these or other options for addressing this point.

3) Methodological clarity – The paper consists of a detailed and thorough methods section. Please check for consistency between the text and Table 1 (and noted by Reviewer 1). Reviewer 2 has also shared a useful suggestion regarding a visualization of the process described on Page 9 (Methodology for Collecting and Evaluating Laws).

4) Table 2 – noted that a graphic designer is working on a revision of this table – if this is now available, please consider submitting with this revision.

5) Contextual information – Reviewer 2 provides several excellent suggestions on clarifying some key points in the introduction, such as how legal instruments and policy are understood in this analysis.

Reviewers' comments:

Reviewer's Responses to Questions

**Comments to the Author**

1. Does this manuscript meet PLOS Global Public Health’s publication criteria? Is the manuscript technically sound, and do the data support the conclusions? The manuscript must describe methodologically and ethically rigorous research with conclusions that are appropriately drawn based on the data presented.

Reviewer #1: Yes

Reviewer #2: Yes

2. Has the statistical analysis been performed appropriately and rigorously?

Reviewer #1: Yes

Reviewer #2: Yes

3. Have the authors made all data underlying the findings in their manuscript fully available (please refer to the Data Availability Statement at the start of the manuscript PDF file)?

Reviewer #1: Yes

Reviewer #2: Yes

4. Is the manuscript presented in an intelligible fashion and written in standard English?

Reviewer #1: Yes

Reviewer #2: Yes

5. Review Comments to the Author

Reviewer #1: Title: Laws for Health and Care Worker Protection and Rights: A study of 178 countries

Thanks to the authors for this thorough and well written paper that documents a significant policy surveillance and cross-national analysis of the national law and policy environment on health and care workers’ protection and rights in (nominally) 178 countries. Impressively, the authors traced over 1200 laws and policies, and analysed their alignment with the ten indicators contained in the Global Health and Care Worker Compact adopted at the 74th WHA in 2021. The study makes a clear contribution to the literature and to global health governance by providing a ‘baseline’ assessment of the de jure policy environment for health worker rights at the broad (national) level. The authors clearly delineate the boundaries of this work which does not extend to evaluation of sub-national policies or assessment of the degree of implementation or enforcement of laws across the 10 indicators. But I largely agree with the authors’ observation that before considering enforcement or implementation, the existence of legal texts is itself an important first step, since such texts help shape how people understand they are to behave and how resources are distributed.

The paper is clearly constructed with good level of detail on the methodology which may not be familiar to many readers. The results are well presented and the Discussion a useful discussion of the findings in context; the trends and implications; and limitations and further research needed. I have just a few comments to strengthen the rigour of the claims made and a few points of clarification.

Abstract

• It would be good to mention in the abstract that the 74th WHA was in 2021 – for readers less familiar with the chronology of WHA.

Methodology

• The detailed description of each indicator area in the Methodology on pages 4-7 is long in the context of a global public health paper. But on the balance I think it is important to retain. It establishes the rationale for indicator selection and serves additionally as a useful primer for readers seeking background on the significance of each indicator area.

• Relatedly, for indicator 6 (pg 6, para 2/3), including key references in support of salaried CHWs (c.f the work of Svea Closser, and Kenneth Maes) is important.

• Table 1 is a useful summary, but doesn’t mention some of the sources (e.g. NATLEX database) from the text (or perhaps slightly different nomenclature used?). Please ensure absolute consistency.

• Thank you for the construction of the www.hcwpolicylab.org library; this is a huge effort and likely to be an important resource. Given the substantive labour it would be good for the currently listed authors of this paper to include ‘group authorship’ (e.g. the HCW policy lab collaborators) documenting a full list of analysts’ names of those who conducted the individual document analysis. This would acknolwedge the significant linguistic expertise required to enable such analysis across a range of different languages (particularly in Asia) and legal and political contexts.

Results

• Overall the results are clear and very well presented.

• One major comment is to recommend the authors more clearly acknowledge and document the proportion of ‘missing data’ for each indicator. Currently (c. pg 10, para 1) the authors note: “for some indicators, no relevant law or policy could be found for some of the countries analyzed or a final determination was not possible. In those cases, we exclude them from our analysis”. However – the authors continue to conduct a comparative analysis of the rates of alignment with each indicator, using summary statistics (%) that imply comparability of the data set for each indicator; when in reality the denominator (total number of countries considered) for each indicator is different. While comparison is still possible it is essential for transparency of this exercise that the different denominator for each indicator is clearly documented at each mention, not just in pie-graph form in Figure 3. So for example on pg 11, para 2 the authors note: “The most aligned area globally is remuneration and fair employment, where 89% of countries have both minimum wage and regulation of working hours for all formally recognized workers.” This sentence does not enable the reader to understand what the 89% refers to. In figure 3, the coverage pie chart shows high level of coverage for that indicator, but the reader must still guess whether the 89% aligned refers to 89% of 150 countries or 160 or 170?

It is extremely important to adjust feature of the results whereever reporting on the % alignment of an indicator, to include something like the bold/starred text below: "The most aligned area globally is the remuneration and fair employment, where 89% of countries **for which there was data (X / X)** have both minimum wage and regulation of working hours for all formally recognized workers.”

Adjustments of this sort should be included for the reporting on every indicator in text (noting mentions that need adjusting in the results AND discussion); and the coverage pie charts in Figure 3 should all be amended to include both total numbers in each category and the summary statistic next to the ‘coverage’ title.

• The challenge of the missing data also adds to some confusion in the opening section of the results (Global Alignment With International Norms…pg 11, para 2, 3). For example, in relation to the statement: 53% of all national laws are fully aligned – what is the denominator here? If the denominator is 178 countries, then presumably full alignment should be adjusted down since the indicator-level assessment reveals missing data in each case. In fact, because of the different denominators (viz number of countries actually able to be evaluated) in each indicator, it isn’t clear to me that this effort to provide a summary analysis across all 10 is appropriate or valid. As it currently stands, Figure 1 inappropriately implies that the proportions of aligned, non-aligned and partially aligned laws applies to all 10 indicators for all 178 countries – when figure 3 shows this can’t be the case. Figure 2 allows more nuance, but again doesn’t reflect the proportion of missing data. Noting here that missing data also tells a story – and transparency in reporting on this is critical to the rigor of this analysis.

• Table 2 while detailed is near impossible to read unless on a large screen with the capacity to zoom in. Some consideration for a format (if at all) appropriate for ‘print publication’ (i.e. the pdf versions) would be good, while retaining the existing version for online use.

Reviewer #2: I would like to congratulate the authors for the lofty goals of this paper. They have done substantial and challenging work to analyse the alignment of legal instruments in 178 countries with State commitments to the Global Health and Care Compact to protect health and care workers. The manuscript has an ambitious scope. I consider it a relevant contribution to the health workforce literature in relation to legislation. However, I have several issues in the paper that would require revisions, particularly to clarify some aspects of their methods/analysis, as well as to link the findings more closely to actionable recommendations. I hope the authors will view my comments as constructive in improving their paper.

Introduction

- Many readers will probably be not familiar with the Compact, which is the starting point for this study. Could you describe the Compact more in terms of its place in international law. Is it an instrument comparable to CEDAW and/or FCTC? Is it an international commitment that is legally non-binding, just like State commitments to the SDGs? Kindly elabore further.

- As many readers of this journal are not lawyers (including myself), I would like the Introduction to include a brief description of the meaning of laws and their hierarchy (e.g., laws vs. circulars vs. guidelines, etc.). This is important because the study examined the legal environment in many countries which included different types of legal instruments and policies.

- The last paragraph of the Introduction needs an explicit articulation of the research objective. The current statement “we have collected national laws and policies of 178 countries to better understand the legal environment within which health and care workers live and work…” sounds more like a method rather than an objective. What were the research objectives/questions that this study sought to answer?

Methods

- What was the basis for having 178 countries? Please clarify. The WHO has 194 Member States.

- The 10 indicators are described without explaining how these were determined. You need to describe the process for identifying them. Was there some form of consultation with stakeholders, review of the literature, or was a workshop held to identify these indicators?

- The indicators for “ensuring occupational health” namely (a) laws to protect health workers and (b) laws to ensure access to protective equipment/supplies seem inadequate. Occupational health has to do also with ensuring a safe and healthy environment through institutionalised programs in the workplace.

- The indicator for “social protection” is the law for parental leaves. This is a very limited indicator. Shouldn’t health insurance coverage for health workers be a measure for social protection? I know firsthand how in some LMICs health workers themselves do not have health insurance coverage (which is an irony).

- Table 1 can be improved by grouping the 10 elements according to the four key domains of the Compact.

- I find the description of the methodology for collecting/evaluating laws to be confusing. In systematic/scoping reviews, there is a PRISMA flowchart that provides a summary of the process for article screening. Although this study is not a systematic review and therefore a PRISMA flowchart does not apply, I was wondering if the authors could visualise their process in a diagram, including showing which databases and websites contributed to what, where consultation with networks of civil society came in, how primary and secondary source documents were differentiated, etc. One attribute of good research is reproducibility and without a clear description of the process for finding laws and evaluating them it would be difficult to replicate this study.

- The HCW Policy Lab (https://www.hcwpolicylab.org) looks interesting. Can the authors describe this website more and whether it is related to this project?

- I would like to suggest that the codebook is provided as part of the Annex.

- I cannot find it in the text of the Methods – what is the criteria for classifying laws as aligned, partially aligned, or non aligned? I think I saw the criteria somewhere in the Annex but this information should be in the Methods.

- “Countries where research did not yield any law or policy answers were classified as no data—reflecting that such a measure could exist but could not be found.” – What about the possibility that there was in fact no such law in the country?

- And how did you account for laws that were written in the non-Latin alphabet (e.g., laws in Chinese characters and thus may not be searchable online), and laws that may only exist offline (i.e. only in print)?

Results

- I think Figure 1 can be excluded as describing the alignment of 1,238 laws as a whole without unpacking the countries is not really policy useful. The information can simply be presented in the text without a need for Figure 1.

- Figure 2 – please place the numbers of countries on the bars in addition to the percentages.

- For the purpose of regional action, I think it would be useful to unpack Figure 2 according to the WHO regions. Action can be taken at the regional levels to address some of the gaps. For example, showing how WPRO compares to SEARO etc. would be a useful analysis, including which countries within each region aligned well (or did not align well) with the Compact.

- Figure 3 – can you put the numbers of countries on the bars and the pie charts? And why are there no partially aligned countries for some indicators (e.g. 3, 6.1/6.2, 8.1/8.2)?

- I think you should also mention and discuss the indicator with the least data (10. whitleblower protections) and most data (6.3 remuneration).

- Table 2 – rather than have the countries in alphabetical order, I think it would be more useful if they are clustered according to WHO regions within the same Table. I note that a graphic designer has been tapped to refine Table 2. The refined version has to go through peer review as well.

- In the last paragraph of Results, there are statistical findings related to country income and country region which seemed out of place as such analyses were not part of the Methods. If you will include these Results here, then the statistical analysis should have been described in the Methods. I saw some of the computations in the Annex but the statistical approach has to be described in the Methods.

- I do not like the word “idiosyncratic” to describe lawmaking in the last paragraph. Can the authors please reformulate.

Discussion

- “Only one country is coded as fully aligned with all 10 indicators.” – Which country was that? This should have been mentioned in the Results. Likewise, was there a country that was least aligned?

- “Notably, 64 countries are aligned with half or more of the indicators studied.” – Is there anything common about these 64 that might explain why they were aligned with the Compact?

- I love the part of the discussion about de jure law.

- Since the starting point of this study was the Compact, I wonder if the authors could also reflect about the role of the Compact in advancing more legislation globally to cover more of the elements of the Compact, including some reflections on State compliance and accountability. The first author of this study had a fine article in the Lancet about State compliance to international commitments which could inform this part of the Discussion.

Conclusion

- The last few sentences of the Conclusion seem anticlimactic and refer to ideas for future research that should be combined instead with the future research section of the Discussion.

Abstract

- Please revise the Abstract in light of the changes to be undertaken above.

Minor comments

- WHO is the funder of this study as well as part of the authorship. A COI should be declared.

- Reference numbers 10, 25, 36, 38, are incomplete.

- For the revised manuscript, please put line numbers at the margins so it would be easier to refer to the text next time.

6. PLOS authors have the option to publish the peer review history of their article (what does this mean?). If published, this will include your full peer review and any attached files.

**Do you want your identity to be public for this peer review?** For information about this choice, including consent withdrawal, please see our Privacy Policy.

Reviewer #1: No

Reviewer #2: **Yes: **Harvy Joy Liwanag

---

## [Decision Letter · Decision Letter 1]

23 Oct 2024

PGPH-D-24-01084R1

Laws for Health and Care Worker Protection and Rights: A study of 182 countries

Dear Dr. Kavanagh,

Thank you for submitting your manuscript to PLOS Global Public Health. After careful consideration, we feel that it has merit but does not fully meet PLOS Global Public Health’s publication criteria as it currently stands. Therefore, we invite you to submit a revised version of the manuscript that addresses the points raised during the review process.

We look forward to receiving your revised manuscript.

Kind regards,

Veena Sriram

Academic Editor

Journal Requirements:

Additional Editor Comments (if provided):

Thank you for the comprehensive efforts in this revision. We are recommending a minor revision based on the reviewer's comments regarding the proportion of countries included in the analysis in the text and - if possible - in Figure 3. I note the comment in your response regarding the difficulty of including numbers within the figure, but one option is to include the denominator under the pie chart on coverage. This will help with the interpretation of the corresponding bar charts. Happy to discuss this further to identify a solution.

Reviewers' comments:

Reviewer's Responses to Questions

**Comments to the Author**

1. If the authors have adequately addressed your comments raised in a previous round of review and you feel that this manuscript is now acceptable for publication, you may indicate that here to bypass the “Comments to the Author” section, enter your conflict of interest statement in the “Confidential to Editor” section, and submit your "Accept" recommendation.

Reviewer #1: All comments have been addressed

2. Does this manuscript meet PLOS Global Public Health’s publication criteria? Is the manuscript technically sound, and do the data support the conclusions? The manuscript must describe methodologically and ethically rigorous research with conclusions that are appropriately drawn based on the data presented.

Reviewer #1: Yes

3. Has the statistical analysis been performed appropriately and rigorously?

Reviewer #1: Yes

4. Have the authors made all data underlying the findings in their manuscript fully available (please refer to the Data Availability Statement at the start of the manuscript PDF file)?

Reviewer #1: Yes

5. Is the manuscript presented in an intelligible fashion and written in standard English?

Reviewer #1: Yes

6. Review Comments to the Author

Reviewer #1: Thanks to the authors for the close attention to the comments and recommendations. Overall these have been addressed and the article reads well, with the exception of 2 outstanding revisions:

* While the authors have added the denominators to the text discussing coverage these remain in an ambiguous format. It is not immediately clear that the (n=xxx) number is in fact a denominator. To make this clearer, it would be very helpful if the format suggested in the original comments were used (see below). This ensures absolute clarity for the reader.

EG: CURRENT: "Just 1% of countries are not aligned (n=172)"

RECOMMENDED FORMAT: "Just 1% (2/172) of countries are not aligned."

[NB: retro engineered the numerator here based on the figures for this example].

* Second, regarding Figure 3, as both reviewers mentioned in the first round it is important to ensure the addition of the denominators to ensure transparency of coverage. Accurate representation of data is as important a consideration as readability, and it its current state Figure 3 still gives a potentially false impression that the coverage of each group is based on an identical number of countries. While the authors have now included this information in the text which is great - it is important for (any) figure to contain all information necessary for a reader to fully understand what is being represented, independent of accompanying text or an appendix. Either by using foot notes, or by adding a x/x figure next to each pie chart, or by other means, this information is critical to the message conveyed by this figure.

7. PLOS authors have the option to publish the peer review history of their article (what does this mean?). If published, this will include your full peer review and any attached files.

**Do you want your identity to be public for this peer review?** For information about this choice, including consent withdrawal, please see our Privacy Policy.

Reviewer #1: No

---

## [Editor Report · Decision Letter 2]

4 Nov 2024

Laws for Health and Care Worker Protection and Rights: A study of 182 countries

PGPH-D-24-01084R2

Dear Dr. Kavanagh,

We are pleased to inform you that your manuscript 'Laws for Health and Care Worker Protection and Rights: A study of 182 countries' has been provisionally accepted for publication in PLOS Global Public Health. 

Best regards,

Veena Sriram

Academic Editor